# Experiences of Frequent Users of Emergency Departments in Health Care Setting in French-Speaking Switzerland: A Qualitative Study

**DOI:** 10.3390/healthcare11091228

**Published:** 2023-04-25

**Authors:** Madison Graells, Luana Schaad, Elodie Schmutz, Joanna Moullin, Olivier Hugli, Jean-Bernard Daeppen, Julia Ambrosetti, Julien Ombelli, Michel Golay, Vincent Ribordy, Patrick Bodenmann, Véronique S. Grazioli

**Affiliations:** 1Department of Vulnerabilities and Social Medicine, Center for Primary Care and Public Health, University of Lausanne, 1011 Lausanne, Switzerland; 2Faculty Health Sciences, School of Population Health, Curtin University, Perth 6845, Australia; 3Emergency Department, Lausanne University Hospital, University of Lausanne, 1011 Lausanne, Switzerland; 4Addiction Medicine, Department of Psychiatry, Lausanne University Hospital, 1011 Lausanne, Switzerland; 5Emergency Department, Geneva University Hospital, 1205 Geneva, Switzerland; 6Emergency Department, North Vaud Hospital Group, 1400 Yverdon-les-Bains, Switzerland; 7Emergency Department, La Broye Hospital, 1530 Payerne, Switzerland; 8Emergency Department, Fribourg Hospital, 1700 Fribourg, Switzerland

**Keywords:** frequent users of emergency department, experience in health care, qualitative inquiry, French-speaking Switzerland

## Abstract

Aims. Frequent users of the emergency department (FUED; five ED visits or more per year) often have negative experiences in health care settings, potentially aggravating their health problems. Scarce research has explored FUED experiences in health care in Europe, none in Switzerland. Thus, this study aimed to conduct an in-depth exploration of FUED experiences in health care settings in Switzerland. Methods. Semi-structured interviews were conducted among 20 FUED (75% female; mean age = 40.6, SD = 12.8). Qualitative data were subject to inductive content analysis. Results. Five main themes emerged from the analysis. The main findings documented that FUED experiences in health care were mostly negative, leading to negative emotions, dissatisfaction and a loss of confidence in the system, although some positive experiences were reported as well. The relationship with health care workers was perceived as playing a key role in FUED experiences. Conclusion. The findings indicate that FUED often have negative experiences in the health care system in Switzerland. The relationship with the health care staff is reported as a decisive ingredient of the experience in health care. Future research is needed to develop awareness-raising interventions for health care staff to improve FUED experiences in health care.

## 1. Introduction

The 2015 Sustainable Development Goals from the United Nations call for actions to achieve universal health coverage by 2030. Universal health coverage means “that all people, everywhere in the world, have access to the high-quality health services they need without facing financial hardship [1]”. Consistent with this agenda, important efforts have been dedicated to decreasing the health inequities typically encountered among socially marginalized populations, such as frequent users of the emergency department (FUED; patients visiting the ED five or more times in the past 12 months) [2]. Driving this high ED use are the often multiple social, medical, psychological and substance use difficulties they typically endure, coupled with frequently limited access to health care outside of the ED [3,4,5,6]. As a matter of fact, ED staff are generally not trained nor equipped to meet the complex needs of this population [7,8]. In response, and with the goal of improving the quality of care and thereby improving health equity for this population, interventions focusing on the FUED’s specific needs, such as case management (CM), have been developed [9]. Based on a holistic socio-medical and nursing assessment, CM reorients FUED to community-based and hospital services and improves the coordination and continuity of care [10]. Increasing evidence shows that CM is a promising intervention to reduce the number of visits to the ED and improves the quality of life among FUED [9,10,11,12,13,14,15] while being positively perceived and experienced by the target population [16,17].

Despite these promising findings, recent international findings highlight that the experience of FUED in the health care system and in the ED remains unpleasant, which may paradoxically contribute to their frequent ED use [18]. Indeed, FUED commonly feel stigmatized, misunderstood and disrespected when they engage with the health care system [18,19,20,21,22]. Correspondingly, FUED are often perceived as illegitimate users of the ED [23], which may contribute to their feeling of discrimination and their dissatisfaction with the health care system. In turn, this may lead to delaying their care, decreasing their compliance, potentially resulting in additional ED consults, as their needs are not fully met at the time of discharge, ultimately worsening health problems [19,20,21,24]. 

Importantly, patient satisfaction is considered to be a crucial indicator of the quality of care [25]. Therefore, these qualitative findings suggest that, despite the important efforts dedicated to improving the quality of care among FUED, much more needs to be done. A possible way to help address this situation is integrating FUED perspectives to design tailored health services. In fact, as highlighted by the WHO, involving people and communities in the development of health services is a key principle when it comes to building high-quality care [1]. Doing so enables the development of interventions tailored to communities’ specific needs. As such, qualitative findings related to FUED experiences in health care may provide important insights into improving the quality of care for this population. However, most of the research exploring FUED experiences with the health care system has been conducted in North America [18]. Additional qualitative explorations involving FUED in a socially and culturally different country, such as Switzerland, is needed to provide key insights into possible specific adjustments to improve the quality of FUED care provided in Switzerland [4,7,10,26]. Such explorations would provide insights for Swiss health care workers concerning FUED experiences. In response, the goal of this qualitative study was to provide an in-depth description of the experiences of FUED in the health care system and in the ED located in French-speaking Switzerland. 

## 2. Materials and Methods

### 2.1. Procedure

This qualitative study was nested in a larger study, which aimed to implement CM in 13 EDs in French-speaking Switzerland while evaluating both the implementation process and the clinical trajectories of the FUED receiving CM interventions [27]. The parent study procedures were developed based on the Generic Implementation Framework (GIF) [28], which includes five stages (i.e., development, exploration, preparation, operation and sustainability). The participants (i.e., FUED agreeing to receive CM and to participate in the study) were recruited during the CM implementation phase from each site included in the parent study. The included participants were invited to complete a longitudinal assessment with the research team by phone to assess their health-related outcomes over 6 months and evaluate their trajectories after receiving CM. In addition, a sub-sample of participants took part in a semi-structured interview, exploring both their experience with the ED and the health care system and their perceptions of CM. The current study focuses on qualitative findings related to their experience with the health care system. The findings related to CM perceptions are reported in a sister paper [17]. A Master-level medical student (L.S.) and a Master-level psychologist (M.G., Madison Graells), under the supervision of a PhD senior researcher (V.S.G.), conducted qualitative interviews by phone. The participants received a voucher of CHF 10 (i.e., about USD 10) in compensation for their time. All of the procedures were approved by the Swiss Ethics Committee on research involving humans of the Canton de Vaud (project number: 2018-0042). All of the participants provided written informed consent.

### 2.2. Participants

Similar to the parent study [27], the patients were eligible if they were 18 years or older, fluent in French and visited the ED 5 times or more in the previous 12 months. Participants were excluded if they had less than 2 vulnerability factors other than frequent ED use (social, somatic, mental health and risk behavior) [8], were unable to provide informed consent, planned to leave Switzerland within 18 months, had a life expectancy of fewer than 18 months, were to be incarcerated or were incarcerated, or had a family member already enrolled in the study. For the current study, given that part of the interview explored their perception of CM (described in the sister study) [17], the lack of CM follow up (less than three appointments with their case manager) and the incapacity to take part in a qualitative interview were added as additional exclusion criteria. In total, 79 FUED were recruited and included across the 8 ED sites implementing CM in the parent study. Of those, the research team randomly selected 40 participants to conduct the qualitative assessment. To perform the random selection of the participants, the research team drew lots for half of the participants included in each ED site included in the parent study. Out of these 40 randomly selected participants invited to participate, 20 accepted, 6 declined, 6 were unreachable by phone or mail, and 8 were excluded (4 because of they were unable to do an interview and 4 because of the lack of follow-up received in their CM). The final sample of 20 participants enabled us to reach data saturation. 

### 2.3. Measure 

Semi-structured interviews were conducted in French with a guide developed by the research team. The guide included open-ended questions and prompts exploring overall FUED experiences with the health care system (e.g., “Tell me about your experience with the health care system?”; “How do you feel when seeking health care?”), interactions and relationships with health care workers (e.g., “How do you find your interactions with health care workers?”), expectations and satisfaction regarding care received (e.g., “To which extent do you feel you are receiving all the help you need at the moment?”; “Why?”). The interviews lasted between 28 and 60 min for a mean duration of 39 min. The Qualitative assessment was conducted over a period of 7 months, from January to July 2020.

### 2.4. Analysis Plan

The interviews were audio recorded and transcribed verbatim. The data were analyzed using an inductive conventional content method [29,30]. A Master-level psychologist (M.G., Madison Graells) and a Master-level medical student (L.S.) independently conducted the initial coding of four different interviews (2 each). They highlighted the ideas emerging from participants’ discourse and created a codebook. In the second step, a PhD senior researcher (V.S.G.) confronted it with two other interviews and provided feedback. The codebook was adapted accordingly during consensus meetings. In the third step, the two researchers (M.G. (Madison Graells), L.S.) who created the codebook double-coded 10% of the interviews independently with the final codebook. Any discrepancies were addressed and discussed until a consensus was found. This process was conducted until an adequate intercoder consistency of more than 80% was obtained [31]. Finally, a single coder (L.S.) coded all the interviews using Atlas.ti version 9.

### 2.5. Researchers’ Characteristics and Reflexivity

The researchers involved in this current study were Caucasian, socially-advantaged clinicians and/or researchers with expertise and/or experience with FUED. They conducted the analysis recognizing that participants and researchers typically construct their own perceptions based on their previous experiences [30]. Their perceptions were not negated but were integrated during the analysis process, in particular during meetings following the initial coding. 

## 3. Results

### 3.1. Sample Description and Qualitative Results

Out of the 20 participants who took part in the qualitative assessment, 75% self-identified as women and 25% as men. The mean age was 40.6 (SD = 12.8). Regarding origin, 90% were Swiss citizens and 10%were European. All of the participants were insured and fully housed. Tertiary school was the most commonly reported highest level of education completed (60%), followed by primary school (35%) and secondary school (5%). Finally, seven participants were recruited from Jura Hospital, four from La Broye Hospital, three from Valais Hospital, three from Fribourg Hospital, two from Neuchâtel Hospital and one from Geneva Hospital. 

Five themes emerged from the inductive qualitative analysis: (1) visiting the health care system: an unpleasant experience; (2) fear, anger, lassitude, loneliness: A loss of confidence in the health care system; (3) disadvantages and limitations of ED; (4) frequent users though … Why? (5) Some positive notes after all.

### 3.2. Visiting the Health Care System: An Unpleasant Experience

Almost all of the participants recalled unpleasant memories of their visits to the ED or other health care settings and even “very, very bad experiences in the hospital” (P1, woman). According to the participants, these experiences were often associated with the attitude of certain health care professionals, who “were a little less human than we would like” (P2, woman) or who “had hurtful words” (P3, woman). One participant explained that she had “ended up crying several times after visiting the ED” (P4, woman).

More specifically, some participants reported that they did not feel heard or understood by the health care professionals they met. For example, one participant described her experiences as: “Very bad … I had a hard time with the doctors, because I felt like they didn’t understand my pain” (P1, woman). Similarly, participants often perceived that their complaints were not taken seriously. For example, Participant 5 (woman) explained that she had to “prove” that she was dizzy or had low blood sugar. Other participants reported encountering caregivers who did not believe them: “some nurses thought that I was faking pain” (P3, women).

Several participants mentioned feeling judged and stigmatized in the health care system or during their ED visits. Some participants felt that it was due to their pathology: “They still stigmatize people who are alcoholics” (P6, woman), “when you’ve had a burn-out, you’re branded” (P7, woman). In addition, some participants felt judged regarding their reasons for visiting the ED: “I’ve been told in the ED that I’m a drug addict coming to get my fix” (P4, woman), while other participants felt judged because they often presented to the ED: “The triage nurse told me I had to stop going to the ED” (P2, woman). 

Finally, a few participants reported unsatisfactory clinical care, explaining that they expected specific treatments or medications and did not receive them: “the doctor on call sent me and asked for this or that test to be done … and they didn’t do them at all” (P7, woman). Conversely, other participants mentioned that they received treatments they did not want or did not tolerate: “there was a medication that I did not want because I could not stand it and some nurses (…) gave it to me anyway and then I was not well” (P8, woman).

### 3.3. Fear, Anger, Lassitude, Loneliness … A Loss of Confidence in the Health System

The negative experiences reported by the participants are not without consequences. Indeed, their need to go to the ED led to fear or stress: “I had this enormous apprehension about the hospital, a very great anxiety” (P6, woman); “I knew I had to go to the hospital because I had these pains and it made me anxious” (P3, woman). A few participants explained that they felt stressed about their ED consultation because they knew they “shouldn’t go” (P9, man), leading to guilt: “I feel guilty the whole time (…) and I’m going to apologize 100 times when I leave the ED” (P2, woman).

Other participants mentioned feeling angry at the health care system: “They didn’t help me with my drinking, because I drank even more. So now (…) I’m going to write to the big boss of psychiatry, because I’m furious” (P6, woman), “When I get to the hospital, I want to beat them up. At the first mistake, I lose my temper. And I call them incompetent; even the doctor I call incompetent” (P10, man); or wearied: “I got a little … a little tired of it to tell you the truth” (P11, woman). A few participants also spoke of their loneliness in navigating the health care system and their impression of being neglected: “I felt left to my own devices” (P7, woman), or even “abandoned” (P12, woman) in certain situations.

As a result of their negative experiences and emotions, the participants commonly explained that they no longer “trust the health care system” because they were “badly oriented” (P6, woman), because the health care management “didn’t lead to anything at all” (P1, woman) or because they considered that health care professionals were “incompetent” (P10, man). In the end, for a few participants, the care provided was considered to be “wasted time and energy” (P9, man) and even if they had follow-up appointments, they felt that they were not adapted to their needs: “Well, let’s say I get help but not what I need” (P9, man).

### 3.4. Experience in the ED: Inconveniences and Limitations 

In addition to the difficulties noted above, participants identified a number of inconveniences in ED setting. One of the most frequently mentioned inconvenience was the waiting time: “When you go to the ED, it’s very crowded; you have to wait a lot. It’s very annoying for us, for the patients” (P7, woman). Beyond the wait, a few participants noted that the uncertainty in which they found themselves was unpleasant: “we wait, we wait, we wait, and we don’t know what’s going on” (P13, woman). 

Another inconvenience raised was the lack of continuity of care from one visit to another. For instance, Participant 4 (woman) explained that when visiting the ED, “it was never the same person (…), everyone had their own interpretation, everyone had their own point of view”. As a result, the ED professionals that they met “always asked the same questions (…) so that was a bit, it was a bit annoying” (P7, woman).

The participants also identified the limitations of the ED setting. For example, one participant explained that for ED professionals, “the possibilities are pretty limited, so other than giving a few medications” they could not do much (P9, man). This is consistent with what others said about going to the ED being “convenient in the moment, for pain relief, when the pain is immediate, but the professionals don’t go looking for the cause of the problem” (P4, woman) or that in the ED, “they don’t have the time to hear your life story” (P13, woman). Finally, Participant 7 (woman) explained that the psychological dimension was often left aside when she received care in the ED, which was problematic because many of her symptoms were related to it: “I also understood (…) that the emergency room (…) is more physical than… and then with me there are really both [the physical and the psychological], it is very mixed the two and it is very strong the two sides”.

### 3.5. Frequent Users Though … Why?

Despite their negative experiences, the participants had repeated consultations with providers in the health care system, especially the ED. This apparent contradiction had several explanations. First, several participants described having a poor quality of life, or a difficult life context in general (family, financial, professional problems, etc.), which could contribute to their need to consult the health system or the ED: “Oh well, I was not doing well, (…) I was a little bit desperate” (P12, woman), “I was also in a period that was not very good” (P14, woman). Many of them suffered from chronic physical health problems: “I’ve had this endometriosis for thirteen years now” (P14, woman) and/or psychological difficulties: “I had depressive symptoms, insomnia” (P9, man). In addition, their health problems were often multiple: “I have several problems that have made my life miserable for the past 6 years” (P7, woman). Many of the participants also suffered from chronic pain: “an atrocious suffering. Every day, I had pain, pain everywhere” (P1, woman). 

Other participants linked their repeated visits to the ED to the fact of being affected with rare and unknown diseases: “The doctors are always looking for what I have” (P2, woman); “it’s something that is not known” (P15, woman). Because their disease was unknown, a few participants explained that it was “very difficult for them to be heard” (P7, woman) in their suffering or to find relief other than by consulting the ED: “After 8 h of vomiting … I am exhausted, the medication resources I have here at home are not enough (…), and so I go to the ED” (P15, woman). 

Some participants reported visiting the ED as they had lost confidence or had issues accessing other parts of the health care system. The ED was perceived as having a reassuring function for patients who no longer trusted their general practitioners or their own ability to manage their health. For example, Participant 18 (woman) mentioned that she “trusted the hospital more than a traditional doctor”. Beyond this reassuring function, a practical side was highlighted. For example, Participant 9 (man) recalled that it could take a long time to see a health professional and that he “didn’t want to wait three weeks for an appointment”. Finally, a few participants sometimes went to the ED on the advice of a doctor: “And he (her cardiologist) told me: “If it doesn’t work, it’s 144 [emergency number in Switzerland]” (P12, woman).

### 3.6. Some Positive Notes after All

In spite of these generally negative experiences, some of the participants recalled positive experiences regarding the health system or in the ED: “I found that we were very well received, in a competent manner” (P12, woman). Several people mentioned that they were generally satisfied with their care: “apart from that [examinations cancelled due to the COVID-19 pandemic], no, I am quite well looked after” (P8, woman) or even totally satisfied: “I’ve always been taken into consideration (…) I’ve always been satisfied with the care” (P15, woman); “I’ve always had a good contact with the health system; I have a great respect for what they do” (P2, woman).

A few participants described health care professionals and teams as particularly competent and caring, enabling positive experiences in the health care system: “in any case, I thank this doctor, because he relieved me a lot, unlike the others” (P16, man); “I have nothing to say about this hospital, on the contrary, I bless them. I find the team extraordinary” (P15, woman). Others noted that they felt “really reassured and confident” (P17, man) when they were in the health care system as well as trust in their health care provider: “this doctor I have great confidence in him”. (P11, woman) Participants also reported feeling taken into consideration: “I felt that they cared for me” (P1, woman). Similarly, participants valued the listening of ED staff about their concern: “And the listening: I had the opportunity to talk to nurses about this feeling of guilt […], they told me, No, no, you come when you feel like you need it, if you don’t feel well, you come [to the ED]” (P12, woman). 

Finally, although less common, a few participants highlighted the positive impact that the care they received had: “you come in very bad, and you leave there, not in great shape, but you are so relieved” (P15, woman).

## 4. Discussion

This qualitative study provides an in-depth exploration of FUED perceptions and experiences with the ED and the health care system in French-speaking Switzerland. Almost all the participants reported unpleasant experiences, balanced by positive ones reported less frequently. The predominantly negative stance concerning health care experiences is consistent with the findings from a recent qualitative systematic review involving the same population [18]. As discussed below, the unpleasant experiences were mainly related to the attitude portrayed by some health care professionals towards them. 

### 4.1. Reasons Leading to Health Care Services Dissatisfaction among FUED in Switzerland

The negative experiences reported by the participants are not without consequences, as they arouse unpleasant emotions (e.g., anger, guilt, anxiety, loneliness) and lead to dissatisfaction with their care and even distrust in the health care system. Previous works have identified five ingredients influencing satisfaction in the ED: timeliness of care, empathy, technical competence, information dispensation, and pain management [32]. However, as highlighted by our results and those of previous studies [19,21,24,33], participants commonly perceived a lack of empathy from professionals, perceived the waiting time as being too long in relation to their problems, received little information at the time of discharge, suffered from chronic pain that was not promptly managed, and perceived certain caregivers as incompetent. This may partially explain the lack of satisfaction with care experienced by FUED, even though previous work documented similar issues in non-FUED as well [34,35].

Specifically to the FUED population, the findings from this study highlight the many inconveniences and limitations of the ED, particularly in the ability to meet the specific needs of the FUED population. These findings are consistent with previous research that revealed that ED staff and settings are typically not equipped to address the complex needs of FUED (e.g., social needs) [7]. As highlighted by a previous ethnographic study conducted in the ED [23], feeling unable to address these patients’ needs may provoke anxiety among health care workers, which may, in turn, lead them to distance themselves from these patients. Importantly, our findings revealed that dissatisfaction in health care was commonly related to negative experiences with health care workers, including the perceptions of not being listened to, not being taken seriously and feeling stigmatized. These findings are congruent with past qualitative research conducted among FUED [33,36] and highlight the importance of providing health care workers with targeted support to help them address FUED complex needs and ultimately improve their relationships with these patients. Doing so may help improve FUED satisfaction in health care, ultimately tackling the quality of care [25]. 

Interestingly, the participants also mentioned several reasons why they visit the ED anyway, such as endorsing complex health needs or seeking reassurance from the ED. In line with these results, two systematic reviews of the literature showed that FUED tend to have more complex health problems than other patients, including, for instance, physical limitations and chronic and mental health problems, which may explain and justify their recurrent passage to ED [37]. Moreover, in addition to the reassuring function of the ED, previous research documented that FUED have a strong attachment to the ED in the sense that it is a familiar place where they are known, recognized, included and considered [18,23]. This could also be an explanation driving FUED to revisit the ED, although this was not raised by the participants in our study.

### 4.2. Highlighting the Weight of the Therapeutic Relationship with Health Care Workers in Health Care Satisfaction among FUED in Switzerland

Interestingly, the findings revealed ambivalent perceptions of the health care system among participants. Indeed, although less common and consistent with past research [18,36], several participants brought up positive perceptions of the health care system, particularly related to encountering caring health care workers or teams or to the quality of the care they received [18,23]. These experiences were related to the attitudes of the health care workers who were competent but also very attentive to the therapeutic relationship. Thanks to the caring and skillfulness of the health care workers, the patients felt well-treated, grateful, satisfied and confident in their care. In fact, these findings are in line with the results from the sister study exploring the experiences of FUED receiving a case management intervention [17]. The latter highlighted that the working relationship with the case manager was perceived as a key ingredient for positive outcomes. 

### 4.3. Drawing Several Recommendations to Improve Quality of Care among FUED

Taken together, the findings highlight the importance of the relationship between FUED and health care workers. Indeed, the majority of the experiences, good or bad, described by the participants were influenced by their perception of the health care workers whom they had or would meet. These findings suggest that improved satisfaction among FUED might be achieved through the establishment of more positive relationships with health care workers, including consideration, compassion, and acceptability. Developing targeted interventions aiming to improve health care staff awareness of FUED characteristics, difficulties and experiences in health care settings while emphasizing the impact of the therapeutic relationship on patients’ experience might represent a means to help achieve this goal. This intervention might be developed through a participatory action research design and might aim to improve staff awareness through FUED testimonies and live exchanges [38]. In addition, given the fact that ED settings are typically not equipped to meet FUED complex needs and acknowledging the many difficulties encountered by health care workers [7,17,39], it may be important to provide them with additional ongoing supervision and coaching to help them navigate the complex situations presented by FUED. Furthermore, besides raising awareness among health care workers, it may be equally important to implement interventions specifically designed to address the complex needs of FUED in ED and primary care settings, such as case management [9,10,17,40]. Doing so may meet health care providers’ needs while providing targeted FUED care. Taken together, these suggestions may help improve the quality of care among FUED. 

### 4.4. Limitations

The findings of this study must be interpreted in light of some limitations. Although data saturation was achieved, this study used a convenience sample of participants included in a larger parent study aiming at implementing CM; the convenience sampling resulted in a small sample size overrepresented by females (75%), whereas previous research, including those conducted in Switzerland, documented a slight tendency towards male preponderance [3,8]. Future research might consider using maximum variation sampling in terms of key FUED characteristics (e.g., gender, age, ethnicity), which would involve a larger sample yet enable to improve FUED representativeness [41]. Next, at the time of the semi-structured interviews, participants had all received or were receiving CM in the larger parent study, which may make this sample different from FUED receiving no CM. That being said, the participants reflected on their previous experience in health care independently upon their experience with CM, and the findings are consistent with past qualitative research conducted among FUED who did not receive CM. Finally, consistent with qualitative inquiry, these findings are specific to the context of the study setting (health care in French-speaking Switzerland) and may, therefore, not be generalizable to the whole of Switzerland. 

## 5. Conclusions

To the best of the authors’ knowledge, this was the first study exploring the experiences of FUED in the health care system in Switzerland. Although some participants recalled a few positive experiences, overall, the findings revealed primarily negative ones, leading to dissatisfaction and a loss of confidence in the health care system. Our results suggested that the relationship between health care professionals and patients plays an important role in FUED experiences of care. Therefore, it seems essential to continue to develop and implement interventions that enable FUED to build trusting relationships with professionals in the health care system, such as CM. In addition, health care workers need to upskill to meet the complex needs of FUED and improve their knowledge of the experiences and difficulties faced by FUED. 

## Data Availability

Not applicable.

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
