# Peer review of "Experiences of Frequent Users of Emergency Departments in Health Care Setting in French-Speaking Switzerland: A Qualitative Study"

_healthcare, 2023, doi:10.3390/healthcare11091228_

Round 1

Reviewer 1 Report

Overall, the introduction provides a clear and concise overview of the research topic and the rationale for the study. The introduction begins by highlighting the increasing number of studies focused on frequent users of the emergency department (FUED) over the past two decades, followed by a definition of FUED and their impact on ED overcrowding. The introduction then describes the multiple social, medical, psychological, and substance use difficulties that FUED typically endure, leading to interventions such as case management to reorient FUED to community-based and hospital services and improve the coordination and continuity of care...

the introduction effectively sets the stage for the study and provides a clear rationale for the research question...

However... the manuscript has serious deficiencies...

1 - The inclusion criteria for the study are not explicitly stated which could be a potential limitation. Additionally, it is not clear how the sample of participants was recruited, which could also be a limitation in terms of the generalizability of the findings. 

2 - The authors do not provide information on how the random selection of participants was performed, which may affect the generalizability of the findings. It would be helpful for future studies to include more details on the randomization process to increase the validity of the sample. 

3 - More critical aspects of the research...

a)       Limited sample size: The study had a relatively small sample size of only 20 participants. This may limit the generalizability of the findings and may not represent the experiences of all FUED in Switzerland.

b)      Qualitative methodology: The study utilized a qualitative methodology, which may not be able to capture the breadth and diversity of FUED experiences. Additionally, the subjective nature of the methodology may introduce bias into the findings.

c)       Limited discussion of previous literature: While the study references previous research on FUED experiences in other countries, there is limited discussion of how these findings compare to the current study's results. A more in-depth comparison could strengthen the review's overall argument.

d)      Lack of specific recommendations: While the review highlights the importance of building trusting relationships between FUED and healthcare professionals, it does not provide specific recommendations for how this could be achieved. More concrete suggestions for interventions or training programs for healthcare professionals could make the review more actionable.

Reviewer 2 Report

This paper describes an study to explore the experiences of frequent users of emergency department (FUED) in health care in Switzerland. The study comprised 20 FUED and the results showed mainly negative experiences, what it seems to be coherent with current literature. The manuscript is well-written, clear, and concise. Introduction is supported by relevant references.

Nevertheless, some flaws have been identified and should be fixed prior to publication:

1. The abstract needs to be improved. It does not explain enough the paper content, the methods or results (e.g., it just names the five themes but does not explain what they means). As it described right now, the conclusion is not supported by the results.

2. In the section Procedure, line 76, the authors claim that the study focuses on FUED experience with the health care system. But it is confusing that those interviewed patients are also being provided with a new case management process.

 3. All the references in the section ‘Analysis plan’ have wrong format and are not included in the reference section.

4. Regarding the five themes extracted from the interviews, there are a relevant set of positive comments (last theme). 7 patients (35% of the sample) had positive experiences, but this is not mentioned in the paper. Title, abstract, and introduction assume that FUED only  have negative experiences. This bias should be fixed in the whole paper.  For instance, line 271 claims “All participants reported unpleasant experiences”, is it true?

5. The sample is small, and the relevance of the results is very limited. The study is developed in a specific context where it has not been carried out previously, but the results do not contribute anything new with respect to what has been published. The main conclusion (line 333 “the relationship between health care professionals and patients plays an important role in FUED’ experience of care”) it only makes sense.

6. If participants in the study were also provided with a case manager (from the bigger study), why they do not include this “personalized care experience” in their answers?  

7. Minor comments:

-          Line 39: delete “ED”

-          Line 144, delete “or”

-          Lines 216-218: quotation marks are missing

Reviewer 3 Report

Very interesting study focusing on the user perspective ‘’ Experiences of Frequent Users of Emergency Departments in Health Care Setting in French-Speaking Switzerland’’ the study discussed the experiences of users and reflecting the emotional part however, it is missing the public health importance of this on the health system and its impact on the access to health services.

Introduction

The introduction needs to reflect the importance of patient satisfaction from the following aspect

·         Universal health coverage

·         Quality improvement of health services

Methodology

Selection of participants: it is not clear, was it random from list of patients? How did it end with 75 % female participants, did you considered the gender balance during the selection?

Results

Elaborate more on the characteristic of the participants, type of illness, for example complaining of chronic or acute diseases, how many of them suffering from mental disorder?

Have you asked about the waiting time? If so, may you reflect the average of waiting time in ED?

Discussion

I would recommend discussing the topic from three different aspects;

1.       patients perspective

2.       Staff perspective

3.       Health system and quality perspective

Round 2

Reviewer 1 Report

I appreciate your efforts in addressing the suggested revisions in your manuscript and submitting the revised version. I have conducted a re-assessment of the manuscript and noted that you have addressed most of the previous comments, which is commendable. 

Overall, the corrections made have improved the quality of the work and made the text more clear and coherent. The modifications made in the sections Abstract, Introduction, Materials and Methods, Results and Discussion were appropriate and have strengthened the argumentation. Additionally, the references have been updated ....

I emphasize the importance of considering sample size in scientific studies to ensure the robustness of the findings. I suggest that the authors discuss possible strategies to increase the sample size or other approaches to mitigate this limitation.

Reviewer 3 Report

no further comments 

Author Response

Dear reviewer, thank you for the re-assessment of our manuscript and for the acknowledgement. We believe that the received comments enabled us to improve our manuscript.